# Venous malformation vessels are improperly specified and hyperproliferative

**Michael J. Schonning**[1¤], **Seung Koh**[1], **Ravi W. Sun**[2], **Gresham T. Richter**[2], **Andrew K. Edwards**[1], **Carrie J. Shawber**[1,3], **June K. Wu**[1]*

**1** Department of Surgery, Columbia University Vagelos College of Physicians and Surgeons, New York, NY, United States of America, **2** Department of Otolaryngology-Head and Neck Surgery, University of Arkansas for Medical Sciences, Little Rock, AR, United States of America, **3** Department of Ob/Gyn, Columbia University Vagelos College of Physicians and Surgeons, New York, NY, United States of America

¤ Current address: Department of Cell Biology and Molecular Medicine, New Jersey Medical School, Rutgers Biomedical and Health Sciences, Newark, NJ, United States of America
* jw92@cumc.columbia.edu

**Data Availability Statement:** All relevant data are within the manuscript and its Supporting Information files.

**Funding:** Funding was provided by DOD W81XWH1910266 (CJS) and DOD

## Abstract

Venous malformations (VMs) are slow-flow malformations of the venous vasculature and are the most common type of vascular malformation with a prevalence of 1%. Germline and somatic mutations have been shown to contribute to VM pathogenesis, but how these mutations affect VM pathobiology is not well understood. The goal of this study was to characterize VM endothelial and mural cell expression by performing a comprehensive expression analysis of VM vasculature. VM specimens (n = 16) were stained for pan-endothelial, arterial, venous, and endothelial progenitor cell proteins; proliferation was assessed with KI67. Endothelial cells in the VM vessels were abnormally orientated and improperly specified, as seen by the misexpression of both arterial and endothelial cell progenitor proteins not observed in control vessels. Consistent with arterialization of the endothelial cells, VM vessels were often surrounded by multiple layers of disorganized mural cells. VM endothelium also had a significant increase in proliferative endothelial cells, which may contribute to the dilated channels seen in VMs. Together the expression analysis indicates that the VM endothelium is misspecified and hyperproliferative, suggesting that VMs are biologically active lesions, consistent with clinical observations of VM progression over time.

## Introduction

The formation of blood vessels initiates with multipotent angioblasts differentiating into venous and arterial endothelial cells (ECs) that form a uniform primary plexus, a process known as vasculogenesis [1]. The plexus is then remodeled and mural cells, pericytes, and vascular smooth muscle cells are recruited to stabilize and mature the blood vessels, a process known as angiogenesis. These processes may be dysregulated in vascular malformations. Vascular malformations are congenital disorders that result in the development of morphologically and architecturally abnormal vascular channels; their development has been suggested to be a result of disruptions in cell fate determination as well as endothelial cell-perivascular cell organization [2, 3]. Venous malformations (VMs) are slow-flow malformations of the venous vasculature [4–6].

W81XWH1910267 (JKW). The funders had no role in the study design, data collection and analysis, decision to publish, or preparation of the manuscript. These studies used the resources of the Herbert Irving Comprehensive Cancer Center Pathology Shared Resources funded in part through Center Grant P30CA013696.

**Competing interests:** The authors have declared that no competing interests exist.

VMs affect up to 1% of the poplation [7, 8], making it the most common vascular malformation. While VMs were originally thought to be biologically quiescent [9], recent research suggests that VMs are biologically active lesions that progress clinically and worsen over time [10–12]. Consistent with their progressive nature, VMs have been linked to genetic mutations that activate the RAS/MAPK (*TIE2, MAP3K3, CCM1*) and PI3K/AKT (*TIE2, PIK3CA*) pathways [13–16], which have a role in regulating EC growth and differentiation [17, 18]. The relationship between these germline/somatic mutations and the VM phenotype are still be elucidated. Both morphologic and expression studies of VMs have pointed to inherent biological and structural differences between vasculature in VM and normal tissues [19, 20]. In extracranial VMs, the pathological endothelium ectopically expressed arterial protein EPHRINB2, suggesting that VM endothelium is arterialized [21]. Similarly, expression of the arterial protein DLL4 was observed in the endothelium in mouse models of cerebral cavernous malformation (CCM), a VM of the brain [22]. Consistent with arterialization of VM vessels, αSMA mural cell coverage is more extensive surrounding the CCM vessels relative to the normal veins in the brain [23]. These findings suggest that VMs are associated with defects in endothelial cell differentiation and dysregulated endothelial cell-perivascular cell interaction. However, these studies are limited in scope and observation.

The goal of our study was to comprehensively analyze expression patterns and vascular morphology in extracranial VM specimens in order to better understand VM pathobiology. We performed a broad investigation of expression patterns of blood endothelial, endothelial progenitor, arterial, and venous proteins, as well as mural cell markers.

## Methods

### Patient recruitment and enrollment

This study was approved by Columbia University's IRB (AAAA9976). Patients with a diagnosis of VM, and scheduled for a resection of their VMs for the patient's direct benefit as part of their clinical care, were recruited by the senior surgeons (JKW and GTR). Informed consent was obtained from either patients or their parents/legal guardians. Patients who were not scheduled to have a surgical resection for their direct benefit were excluded. Excess VM tissues not needed for submission to the Department of Pathology were taken to the laboratory to be processed. Patients were recruited between June 2007 through March 2018. Control neonatal foreskins were anonymously collected and no patients (or parents/legal guardian) signed a consent form. The waiver for consent was approved for controls tissues in the same IRB.

### VM specimens and controls

VM diagnosis was confirmed with clinical examination as a lesion with blue hue discoloration that may engorge in the dependent position or with Valsalva maneuvers, imaging studies demonstrating absence of fast-flow, and/or clinical pathology reports confirming dilated vascular channels with negative D2-40 (PODOPLANIN) staining. All tissues were fixed in 10% formalin and dehydrated with alcohol before being paraffin-embedded. Fetal tissue array consisting of 78 cores from 26 male and female fetuses at 5 months of gestation was used for fetal tissue expression analysis (BE01014; Tissue Biomax).

### Immunohistochemistry

Serial sections (5μm) were deparaffinized, rehydrated, and blocked as previously described [24]. Primary antibodies (S1 Table) were detected with Alexa Fluor-conjugated Donkey-secondary antibodies at 1:1000 (Invitrogen) and slides mounted with DAPI Mounting Media

(Vector). Specificity of NOTCH3 antibody was determined by preblocking with a NOTCH3--specific blocking peptide at a ratio of 5:1 NOTCH3 blocking peptide to antibody. Images were captured on Zeiss AxioCam MRc camera with Zeiss Zen software at 20X or 40X magnification. All images for each staining were processed in the same manner.

## Expression analyses

Protein expression in VMs was compared to control neonatal dermis. For each staining, 2–5 images were analyzed per tissue section and 3–8 vessels from each field of view were selected as the region of interest (ROI) for quantification. Protein expression was quantified as signal intensity for VEGFR2, EPHB4, DLL4, EPHRINB2, and CD146 over the perimeter or length of endothelium (intensity/μm) of unadjusted images using ImageJ (NIH). VECADHERIN, CD31, COUP-TFII, NOTCH3, PDGFRβ, CD133, and CKIT expression was measured as a percentage of positive ECs/total ECs in selected ROIs and averaged for each VM specimen. αSMA expression was tabulated as a descriptive phenotype based on arterial or venous characteristics. Arterial phenotype was defined as continuous, multi-layer mural cells surrounding ECs, whereas venous phenotype was defined as a continuous single layer, or lack of mural cells. Disrupted or discontinuous mural cell coverage was defined as a "discontinuous" phenotype. Proliferation was assessed by counting the number of KI67+/VECADHERIN+/DAPI + nuclei and dividing by total # ECs (VECADHERIN+/DAPI+ cells) over 5 high powered fields (hpfs; 20X). Average EC length was determined as the length of the perimeter of each vessel divided by the # of ECs (VECADHERIN+/DAPI+ cells). Statistical analysis was performed using Graphpad. Unpaired student's *t-test* was used to compare control neonatal dermis to VM specimens and a p value of <0.05 was considered statistically significant. Statistical analysis was performed with one-way ANOVA for DLL4 and EPHRINB2 and post-hoc *t-test* was used for comparison between groups.

## Results

### Tissue characteristics

VMs analyzed (n = 16) were extracranial in the subcutaneous soft tissue and located in various anatomic locations, including the head and neck (n = 7), trunk (n = 2), upper extremity (n = 3), and lower extremity (n = 4). Neonatal foreskin (n = 5) served as normal control dermal tissue.

### VM endothelium has reduced and mislocalized expression of essential EC adhesion proteins

Morphological analysis of VM tissues revealed that pathological vessels were dilated, irregularly-shaped vascular channels lined by disorganized endothelial cells with nuclei often oriented perpendicular to the lumen, suggesting defects in EC polarity or cell-cell associations (S1 Fig). To evaluate the tight and adherens junctions, VM specimens were stained for VECADHERIN, CD31, and VEGFR2.

Expression of VECADHERIN and CD31 was analyzed across multiple stainings to achieve a total of n = 16 VMs. VECADHERIN expression was punctated and discontinuous relative to neonatal dermal vessels in 15/16 VMs evaluated (Fig 1A) and there were significantly fewer VECADHERIN+ ECs lining VM vessels than in control tissues (Fig 1C). Similar to VECADHERIN, CD31 expression was not continuous, with a loss of CD31 expression observed in 15/16 VMs (Fig 1A and 1B) and significantly less CD31+ ECs lining the VM endothelium than found in controls (Fig 1D). Unlike VECADHERIN and CD31, VEGFR2, a pan-endothelial

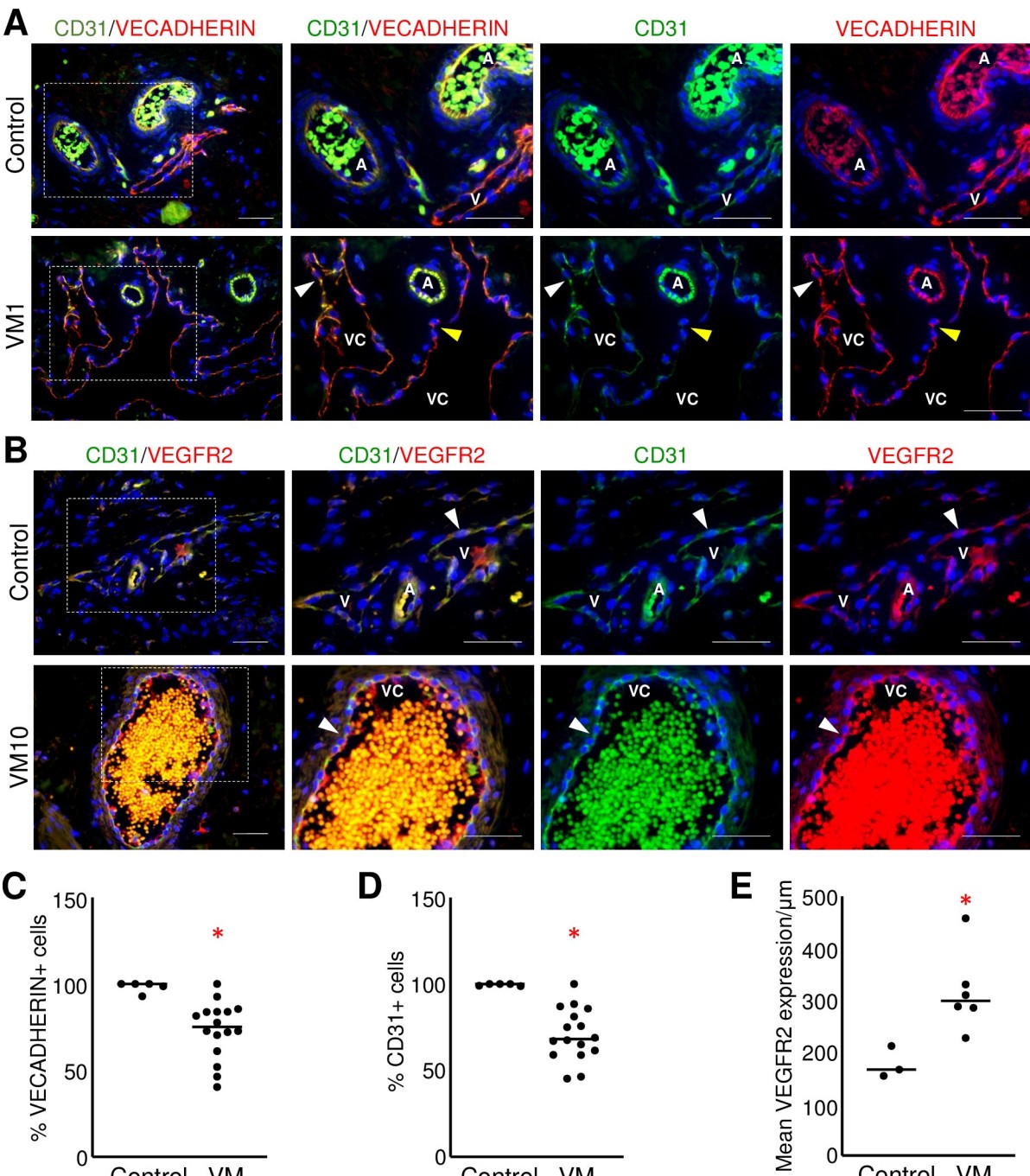

**Fig 1. CD31, VECADHERIN, and VEGFR2 expression was altered in the VM endothelium.** (A) Representative images of VM and control neonatal dermis co-stained for CD31 and VECADHERIN. White arrowheads mark discontinuous CD31 and VECADHERIN expression observed in VM vessels. Yellow arrowheads mark CD31-/VECADHERIN+ ECs. (B) Representative images of VMs and control neonatal dermis co-stained for CD31 and VEGFR2. White arrowheads mark CD31+/VEGFR2+ ECs. (A, B) Boxed areas are enlarged to the right. Scale bars—50μm. A-artery, V-vein, VC-VM channel. (C) Quantification of percent VECADHERIN+ ECs in VMs (n = 16) and controls (n = 5). Bar represents median value, *p<0.005. D) Quantification of percent CD31+ ECs in VMs (n = 16) and controls (n = 5). Bar represents median value, *p<0.001. (E) Mean VEGFR2 expression determined as signal intensity normalized by vessel length in VMs (n = 6) and controls (n = 3) tissues. Bar represents median value, *p<0.05.

marker, had significantly higher signal intensity in VM endothelium than in controls (Fig 1B and 1E). This high expression of VEGFR2 is consistent with ECs lining VM channels.

## VMs have characteristics of arterial vessels

Since venous ECs in VMs have been shown to express the arterial marker EPHRINB2 [21, 22], we next characterized VM endothelium by staining for both venous (COUP-TFII, EPHB4) (Fig 2) and arterial (DLL4, EPHRINB2) (Fig 3) EC proteins. VM endothelium expressed both venous proteins, but their expression differed from that of the venous endothelium in controls (Fig 2A and 2B). The percentage of COUP-TFII+ cells in VM endothelium was significantly less than control veins (Fig 2A and 2C). By contrast, EPHB4 expression was significantly increased in VM ECs (Fig 2B and 2D) relative to controls.

VM endothelium also misexpressed arterial proteins. In control tissues, DLL4 expression in the venous endothelium was significantly less than that of arterial ECs (Fig 3A and 3C). In contrast, DLL4 expression in VM endothelium was significantly increased relative to control veins, and was similar to DLL4 expression observed in control arteries (Fig 3A and 3C). EPHRINB2 expression in VMs was more similar to arteries than veins but the difference was not significant (Fig 3B and 3D). The presence of arterial proteins had no correlation with the status of venous protein expression in the VMs.

Given that VM endothelium expressed arterial EC proteins, we assessed the mural cell investment and organization of VM vessels. Mural cell coverage differs between arteries and veins [25]. Arteries are surrounded by several layers of αSMA, PDGFRβ, and NOTCH3 expressing mural cells, while veins are more sparsely covered with a single layer of PDGFRβ and NOTCH3 expressing mural cells [2, 25–27]. In control tissues, arterial vessels were invested with a continuous, multi-layer of αSMA+ and NOTCH3+ mural cells, whereas the veins had sparse or no mural cell coverage (Fig 4A and 4B). Only a minority of VM channels exhibited a vein-like mural cell phenotype (22%) of either sparse or no mural cell coverage (Fig 4C). While 8% of VM channels were surrounded by continuous, multi-layer mural cells, similar to arteries, the majority of VM channels (70%) were surrounded by discontinuous multi-layer mural cells (Fig 4C). No association between channel-size and perivascular cell coverage was observed in VM specimens. Thus, misexpression of arterial EC proteins in the endothelium of VMs was associated with increased and disorganized perivascular coverage, suggesting the VM endothelium failed to properly differentiate and displays characteristics of both arteries and veins.

## VM endothelium expressed NOTCH3 and PDGFRβ

Unlike the control tissues where NOTCH3 and PDGFRβ expression was restricted to perivascular cells surrounding the control arteries, their expression was also observed in VM endothelium (Fig 5A, S2 Fig). NOTCH3 expression was analyzed over 2 independent experiments to achieve a total of 11 VM specimens. The number of NOTCH3+ ECs in VM endothelium was significantly higher than in control tissues (Fig 5B). Similar to NOTCH3, spotty endothelial PDGFRβ expression was observed in VM endothelium, which was significantly higher than in control vessel ECs (Fig 5C, S2 Fig). Since NOTCH1, 2, and 4 are also expressed in ECs, we confirmed the specificity of the anti-NOTCH3 antibody using a NOTCH3 blocking peptide (S3 Fig).

We previously observed that NOTCH3 is expressed in the immature ECs of infantile hemangiomas (IH), a blood vascular anomaly [28]. A human fetal tissue array was used to further explore NOTCH3 expression in immature endothelium, and endothelial NOTCH3 expression was determined. NOTCH3 was expressed in at least 20% of ECs in 14/19 fetal organs queried

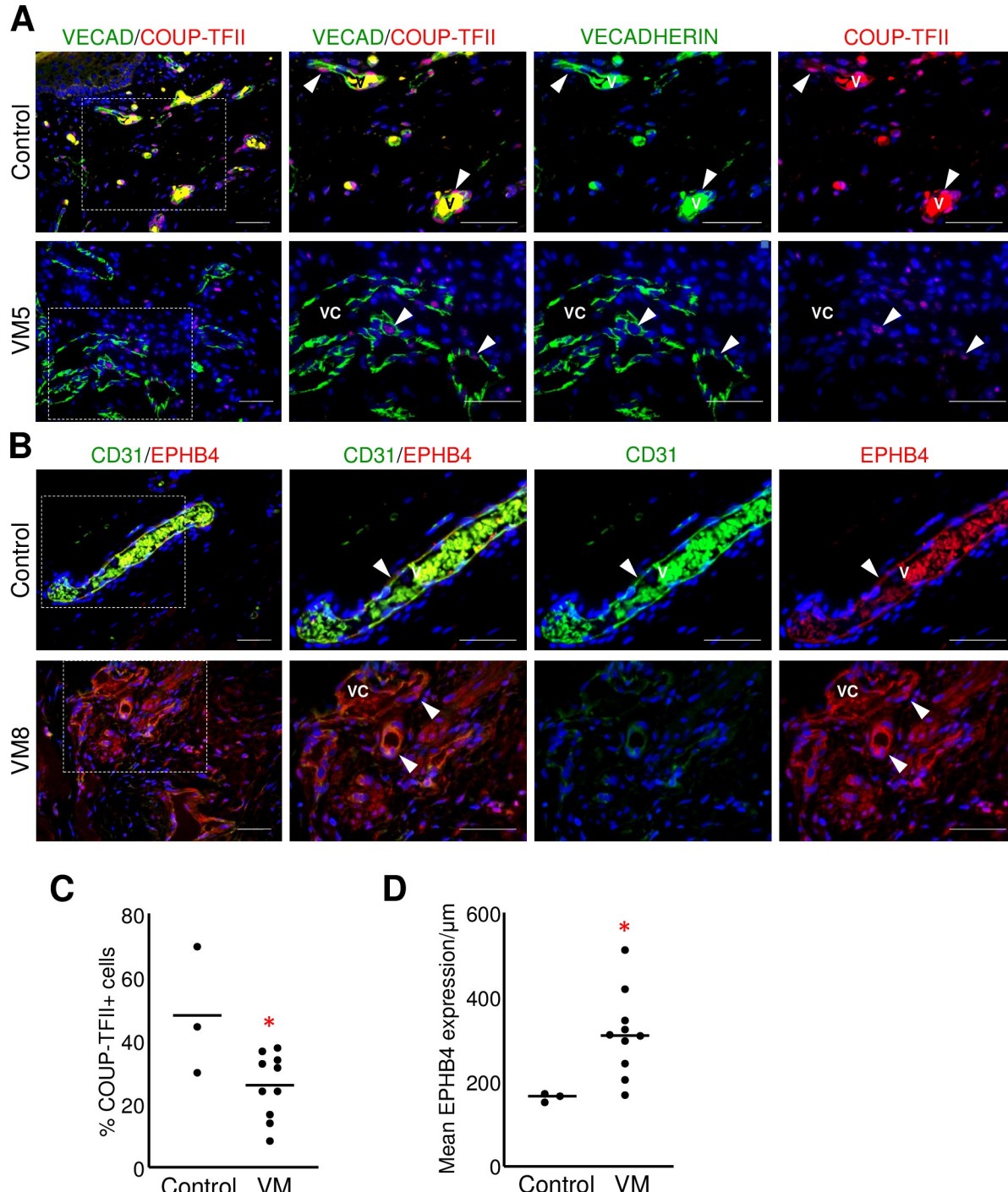

**Fig 2. Venous endothelial proteins were altered in VM endothelium.** (A) Representative images of VMs and control neonatal dermis co-stained for VECADHERIN (VECAD) and COUP-TFII. White arrowheads mark VECADHERIN+/COUP-TFII+ ECs. (B) Representative sections of VMs and control neonatal dermis co-stained for VECADHERIN and EPHB4. White arrowheads mark CD31+/EPHB4+ cells. (A, B) Boxed areas are enlarged to the right. V-vein, VC-VM channel. Scale bars—50μm. V-vein, VC-VM channel. (C) Percent COUP-TFII + ECs in VMs (n = 10) and controls (n = 3). (D) Mean EPHB4 expression normalized by vessel length in VMs (n = 10) and controls (n = 3). (C, D) Bars represent median values, *p<0.05.

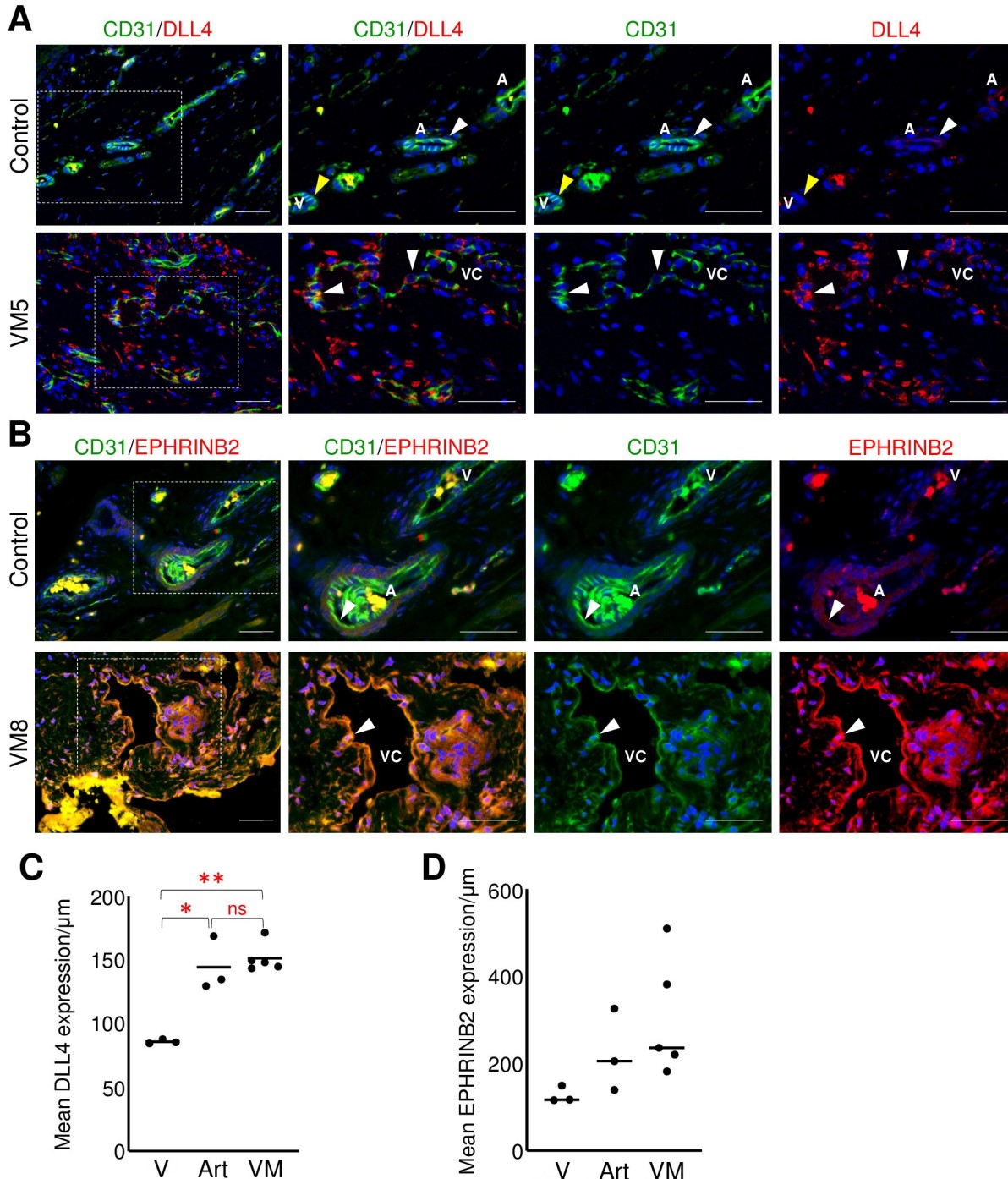

**Fig 3. VM endothelium misexpressed arterial EC proteins.** (A) Representative images of VM and control neonatal dermis co-stained for CD31 and DLL4. White arrowheads mark CD31+/DLL4+ ECs. Yellow arrowheads mark CD31+/DLL4- ECs. (B) Representative images of VM and control neonatal dermis co-stained for CD31 and EPHRINB2. White arrowheads mark CD31+/EPHRINB2+ cells. (A, B) Boxed areas are enlarged to the right. A-artery, V-vein, VC-VM channel. Scale bars– 50μm. (C). Mean DLL4 expression normalized by vessel length in VMs (n = 5) and controls (n = 3). Bar represents median value; ANOVA, p<0.0005; T-Test *p< 0.01, **p<0.0001, ns, non-significant. (D) Mean EPHRINB2 expression normalized by vessel length in VMs (n = 5) and controls (n = 3). Bar represents median value. V-vein, Art-artery, VM-venous malformation.

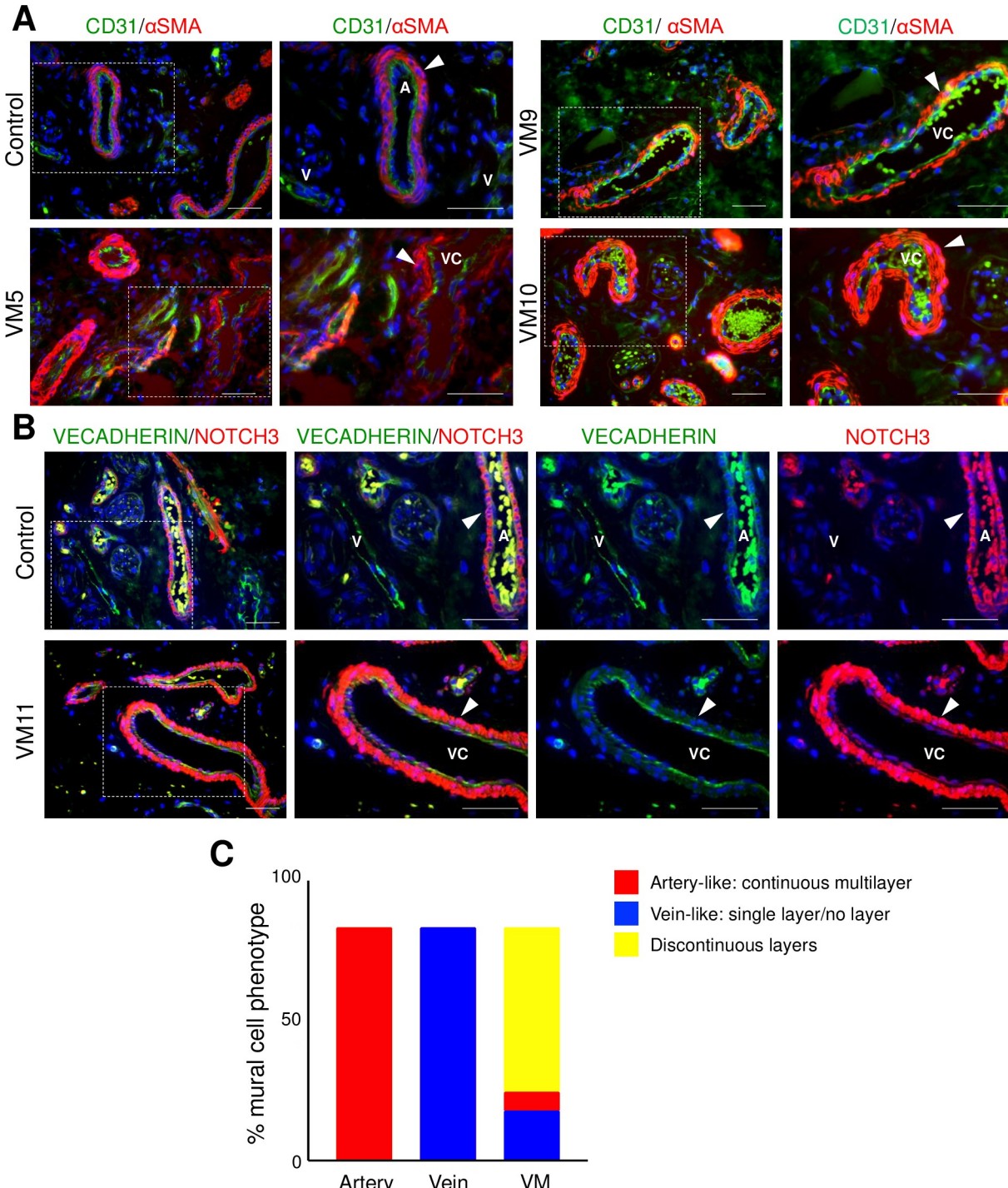

**Fig 4. VMs had increased and disorganized αSMA expressing mural cells.** (A) Representative images of VM and control neonatal dermis co-stained for CD31 and αSMA. White arrowheads mark CD31-/αSMA+ mural cells. (B) Representative sections of VMs and control neonatal dermis co-stained for VECADHERIN and NOTCH3. White arrowheads mark VECADHERIN-/NOTCH3+ mural cells. (A, B) Boxed areas are enlarged to the right. A-artery, V-vein, VC-VM channel. Scale bars– 50μm. (C) Percentage of mural cell phenotype (arterial-like, continuous multi layer; vein-like, single layer/no layer, and discontinuous layers) in arteries and veins of controls (n = 3), and VMs (n = 11).

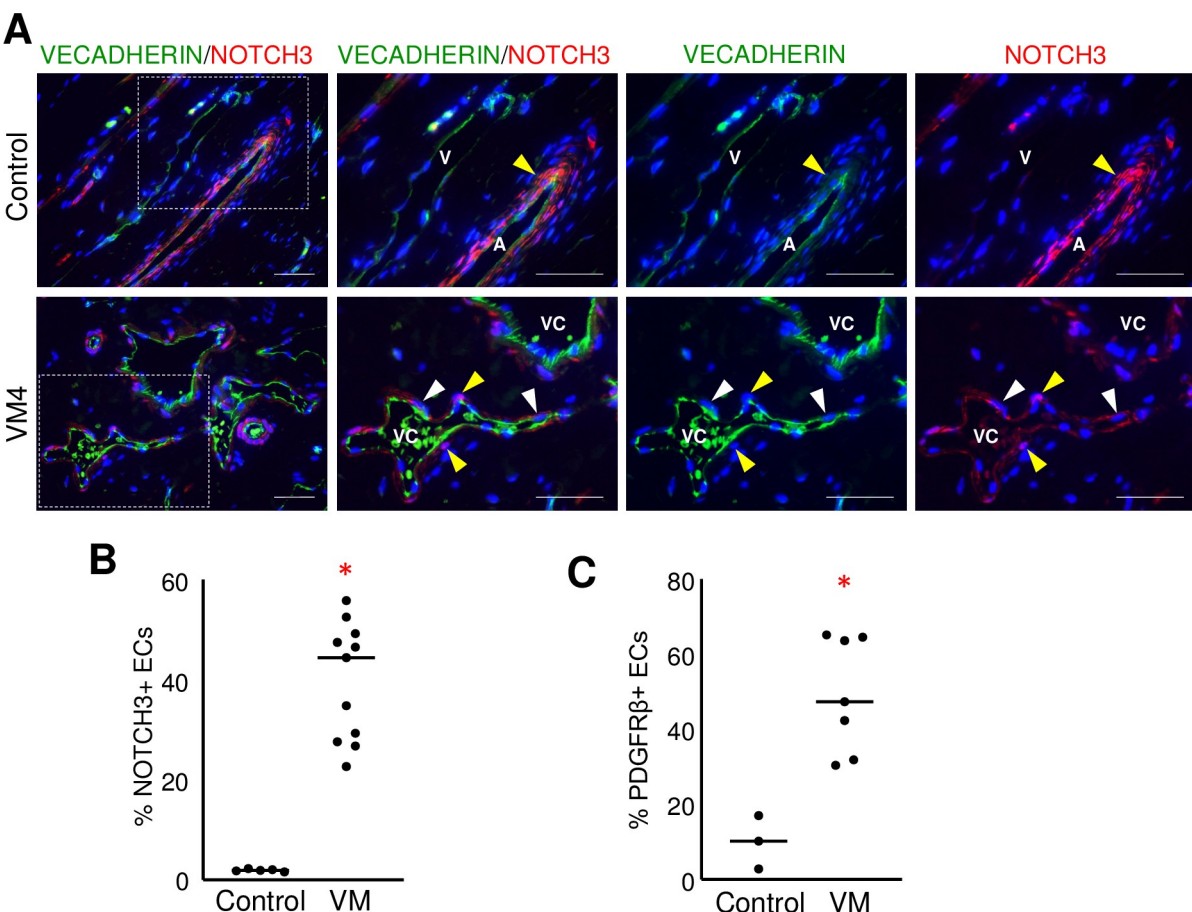

**Fig 5. NOTCH3 and PDGFRβ were expressed in the endothelium of VMs.** (A) Representative images of VMs and control neonatal dermis co-stained for VECADHERIN and NOTCH3. Yellow arrowheads mark VECADHERIN-/NOTCH3+ mural cells, and white arrowheads mark VECADHERIN+/NOTCH3+ ECs. Boxed areas are enlarged to the right. A-artery, V-vein, VC-VM channel. Scale bars—50μm. (B) Quantification of percent NOTCH3+ ECs in VMs (n = 11) and controls (n = 5). Bar represents median value, *p<0.0001. (C) Quantification of percent PDGFRβ+ ECs in VMs (n = 7) and controls (n = 3). Bar represents median value, *p<0.005.

(S2 Table, S4 Fig). Thus, similar to the endothelium in VMs and in IHs, the immature fetal endothelium expressed NOTCH3; this was not observed in the more mature postnatal vessels (Fig 5A).

## VM endothelium expressed EC progenitor proteins

Since VM endothelium expressed NOTCH3 (similar to fetal tissues), we next determined the expression of endothelial progenitor proteins CD133, CKIT, and CD146 in VM and control tissues. In the neonatal dermis, an occasional CD133+/VECADHERIN+ EC was observed within the endothelium of blood vessels (Fig 6A). By contrast, there was a significantly higher percentage of CD133+ ECs in VM vessels relative to control tissues (Fig 6C). CKIT expression was not seen in control tissues; in contrast, a subset of CKIT+ ECs were found in VM endothelium (Fig 6B and 6D). CD146 expression was similar between VM and control vessels (Fig 6E, S5 Fig). However, CD146 expression was inconsistent in VM vessels with areas of both intense and faint CD146 staining and other areas with no CD146 expression. This data demonstrates that VM endothelium had increased or altered expression of EC progenitor proteins when compared to neonatal dermal vessels, consistent with the immaturity of VM endothelium.

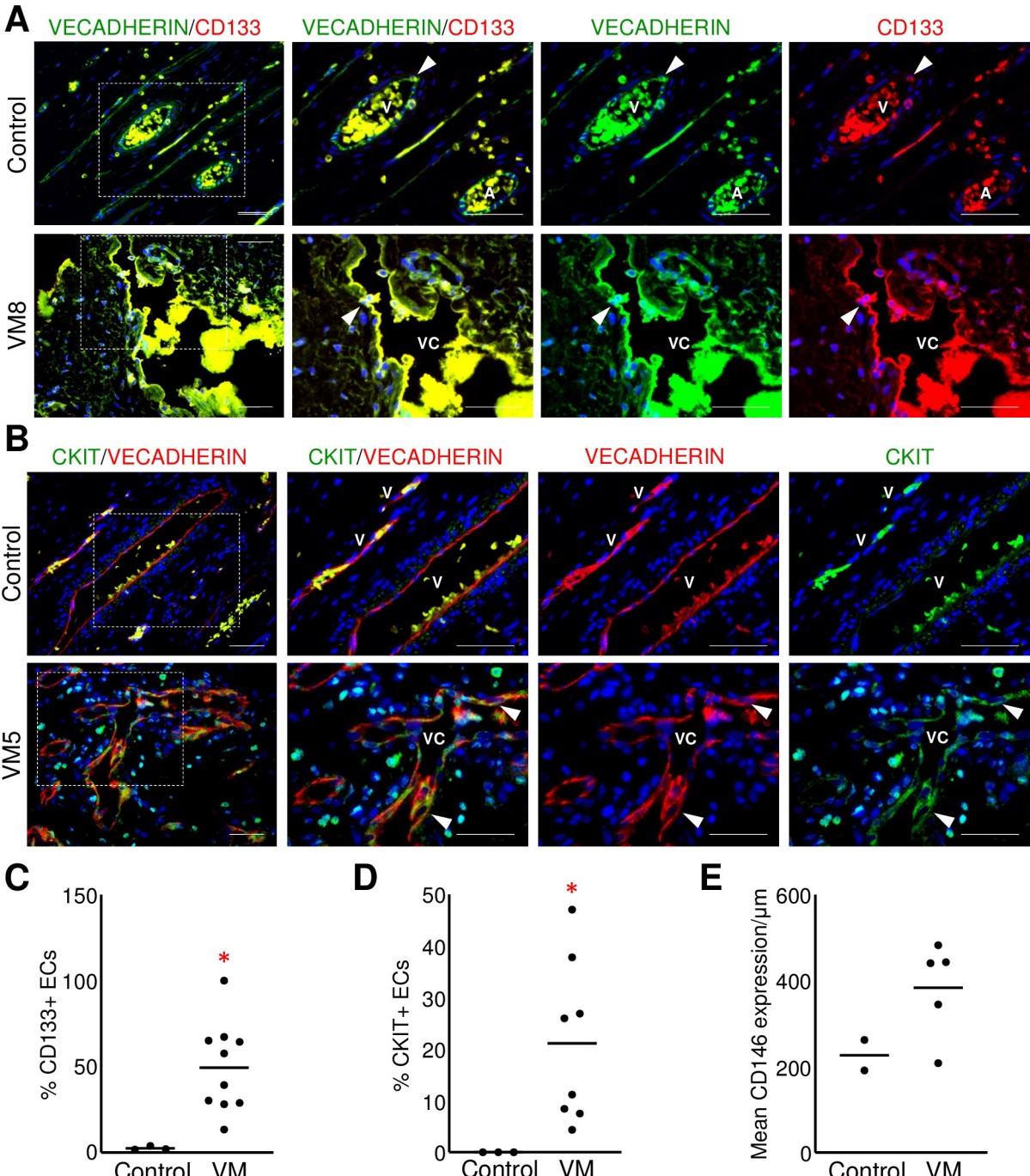

**Fig 6. VM endothelium expressed progenitor markers.** (A) Representative images of VM and control neonatal dermis co-stained for VECADHERIN and CD133. White arrowheads mark VECADHERIN+/CD133+ ECs. (B). Representative images of VM and control neonatal dermis co-stained for VECADHERIN and CKIT. White arrowheads mark VECADHERIN+/CKIT+ ECs. (A, B) Boxed areas are enlarged to the right. A-artery, V-vein, VC-VM channel. Scale bars—50μm. (C) Quantification of percent CD133+ ECs in VMs (n = 10) and controls (n = 3). Bar represents median value, *p<0.02. (D) Quantification of percent CKIT+ ECs in VMs (n = 8) and controls (n = 3). Bar represents median value, *p = 0.05. (E) Mean CD146 expression normalized by vessel length in VMs (n = 5) and controls (n = 2). Bar represents median value.

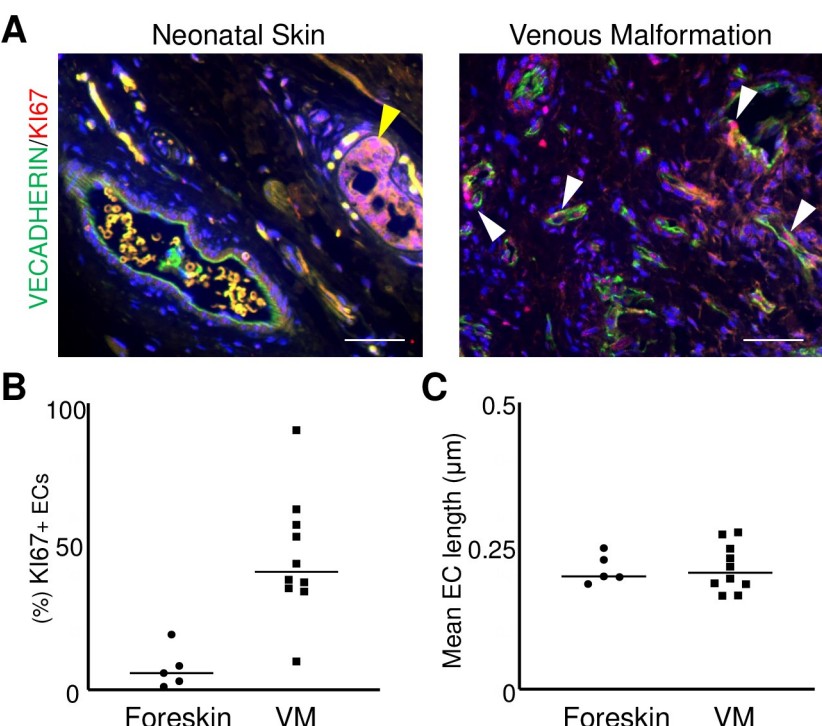

**Fig 7. Increased in proliferative ECs in VM vessels.** (A) Representative images of VM and control neonatal dermis co-stained for VECADHERIN and KI67. White arrowheads mark VECADHERIN+/KI67+ ECs. Yellow arrowhead marks a KI67+ non-EC in the control. Scale bars—50μm. (B) Quantification of percent KI67+ ECs in VMs (n = 10) and controls (n = 5). Bar represents median value, *p<0.0002. (C) Quantification of EC length in VMs (n = 10) and controls (n = 5). Bar represents median value.

## VM endothelium were hyperproliferative

Since immature endothelium has increased proliferation, we stained VM specimens (n = 10) for the proliferative marker KI67 with the EC protein VECADHERIN. Relative to control neonatal dermal ECs (n = 5), the VM endothelium had a significant increase in the number of KI67+/VECADHERIN+ ECs (Fig 7A and 7B). Average EC length, a proxy for EC size, did not differ between control and VM specimens (Fig 7C). These results demonstrate that VM endothelium is hyperproliferative and not hypertrophic.

## Discussion

Expression studies, presented here, demonstrate that the VM endothelium was disorganized, proliferative, and partially arterialized, while also expressing venous EC proteins. ECs in VMs expressed significantly higher levels of VEGFR2, a protein expressed in endothelial progenitor cells and differentiated ECs, thus confirming the endothelial identity of cells lining VM channels. In contrast, VM vessels had significantly reduced expression of key endothelial junctional proteins CD31 and VECADHERIN, and their expression patterns were spotty. Abnormal VECADHERIN and CD31 expression at EC-EC junctions in VMs may lead to a failure of EC polarization and contribute to the altered EC morphology in VMs. VM endothelium also had a significant increase in EC proliferation, significantly upregulated expression of the arterial protein DLL4, and were surrounded by disorganized PDGFRβ+/NOTCH3+/αSMA+ mural cells. While the mural cells were more similar to arteries than veins, they were often disorganized and discontinuous. Finally, VM endothelium had significantly higher expression of

endothelial progenitor cell proteins CKIT and CD133. Taken together, this data demonstrates that the VM endothelium failed to properly differentiate, co-expressed both venous and arterial EC proteins, and was immature and hyperproliferative. These expression patterns suggest a failure in proper venous EC differentiation.

We observed that a majority of VMs had reduced and/or inconsistent expression of the tight and adherens junction proteins, CD31 and VECADHERIN. Consistent with these findings, loss-of-function *Krit1* mutations in CCM1, an intracranial VM, has been associated with reduced expression of VECADHERIN at the cell surface [29]. In VMs, the decrease in CD31 and mislocations of VECADHERIN may contribute to the altered EC morphology, increased permeability, and thrombi formation [30–32]. In ECs, CD31 functions as a flow sensor and promotes EC polarity via VECADHERIN regulation [33–35]. The altered VMEC morphology suggested a loss of cell polarity that could be secondary to a loss of EC flow sensing. While disrupted EC junctions in VM endothelium may contribute to the morphological changes in the ECs, CD31 and VECADHERIN are also required for vessel barrier formation. This loss of barrier function would allow extravasation of blood into the surrounding tissues, which is often experienced by VM patients. Finally, the reduction of CD31 may contribute to the intravascular coagulopathies seen in VMs as there is an increase in thrombi in *Cd31* null mice [30].

As expected, VM endothelium expressed venous proteins COUP-TFII and EPHB4, although at different levels when compared to normal veins. The VM endothelium also expressed the arterial protein DLL4 and were surrounded by disorganized PDGFRβ+/αSMA +/NOTCH3+ perivascular cells (Fig 4) which suggests that the VM endothelium has partially arterialized. A similar phenotype has been reported for both intracranial and extracranial VMs, suggesting that this is a common pathological phenotype observed in VMs [5, 10, 21, 22, 36–38]. Unlike arteriovenous malformations (AVMs), where veins are exposed to arterial flow due to the absence of an intervening capillary bed, VMs are slow-flow lesions. Given this, altered flow is most likely not promoting the expression of arterial proteins in VMs. Additionally, arterialization is not likely downstream of PI3K/AKT signal hyperactivation. In mice, ectopic EC expression of a *Pik3ca* variant associated with VMs was associated with venous specification of the vasculature characterized by a reduction in pericyte coverage and downregulated EPHRINB2 expression [39].

The arterialization of the VM endothelium may be secondary to either loss of VECADHERIN regulation of VEGFR2 or endothelial NOTCH3 signaling. In the endothelium, VECADHERIN suppresses VEGFR2 and its loss leads to an increase in VEGFR2 signaling [31]. The level of VEGFA/VEGFR2 signaling promotes distinct EC fates: high VEGFA leads to arterial ECs, while low levels of VEGFA promotes venous ECs [40]. Thus, VECADHERIN$^{low}$/VEGFR2$^{high}$ ECs (Fig 2) in VMs may lead to hyperactivation of the VEGFA/VEGFR2 pathway, which then induces the expression of arterial proteins. Alternatively, ectopic expression of NOTCH3 in ECs was associated with both increased DLL4 expression and increased tumor angiogenesis in a murine xenograft model [41].

Consistent with Diehl et al. [21], we demonstrate that the VM endothelium has a significant increase in EC proliferation with no significant changes in EC size relative to the neonatal vasculature (Fig 7), suggesting that VMs are hyperproliferative, and not hypertrophic, lesions. This hyperproliferation is most likely due to hyperactivation of the PI3K/AKT pathway. Inhibiting the PI3K/AKT pathway in VMs carrying activating mutations suppressed EC growth in mice [39, 42]. In transgenic mice, ectopic expression of a *Pik3ca* variant in ECs led to increased EC proliferation and a hyperplastic retinal vascular plexus [39]. These expression studies suggest that alternative pathological mechanisms could also contribute to the proliferative phenotype in the VM endothelium. The increase in EC proliferation may be due to loss of VECADHERIN and its function as a suppressor of VEGFR2 signaling, which is pro-

proliferative in ECs. Alternatively, EC NOTCH3 functions may contribute to EC proliferation in VMs. Unlike the neonatal vasculature, we observed NOTCH3 expression in the biologically active endothelium in a majority of human fetal organs, while EC NOTCH3 expression has been reported for tumor vessels [41]. Consistent with the murine model, we have observed NOTCH3 expression in the proliferative ECs in IHs [24, 28]. As NOTCH3 signaling upregulates PDGFRβ in perivascular cells [26, 43], NOTCH3 may also regulate vascular PDGFRβ in the ECs to promote proliferation. In ECs, PDGFRβ expression is associated with proliferative angiogenic ECs. Blocking PDGFRβ in cultured ECs blunted angiogenic responses; PDGFRβ activity loss results in increased endothelial cell numbers and capillary caliber in vitro [44] as well as endothelial maturation and patterning defects in vivo [45]. Taken together there may be three pathways that converge to promote EC proliferation in VMs: 1) PI3K hyperactivation, 2) loss of VEGFR2 suppression due to VECADHERIN defects, and 3) the NOTCH3/PDGFRβ signaling axis.

Together, misexpression of arterial proteins and increased proliferation suggest that the endothelium in VMs has failed to terminally differentiate. Consistent with this idea, the VM endothelium had significantly higher expression of endothelial progenitor cell markers CD133 and CKIT. Although PDGFRβ is a mural cell marker in the vasculature [25, 46], it is also expressed in circulating endothelial progenitors, hemangioblasts [47, 48], and the immature proliferative endothelium of infantile hemangiomas [24]. This data further supports the theory that VM endothelium retained progenitor identity due to incomplete or incorrect differentiation and specification. How the misspecification and persistent expression of progenitor markers affects VM pathobiology remains to be elucidated.

This study has limitations. As an expression study, it is observational and does not provide insight into mechanisms of VM pathogenesis. It is not known whether or how genetic mutations seen in VMs contributed to the expression patterns described here. It is also unknown whether these protein misexpression patterns are the cause or result of VM pathobiology. Future studies on endothelial biology of VM-dervied endothelial cells may provide further insight.

## Conclusions

We found that VM vessels failed to properly differentiate into venous ECs, misexpressed arterial and endothelial progenitor proteins, had increased and disorganized mural cell coverage, and were hyperproliferative. These findings suggest that VM endothelium and ECs differ from normal venous endothelium, are not as biologically quiescent as previously thought, and are consistent with clinical observations of a progressive natural history [11].

## Supporting information

**S1 Fig. Altered endothelial cell morphology in VMs.** Representative H&Es of VMs and control neonatal skin. Boxed areas are enlarged to the right. Blue arrowheads highlight normal EC morphology. Black arrowheads mark ECs with abnormal morphology. A-artery, V-vein, VC-VM channel. Scale bars—50μm.
(TIF)

**S2 Fig. VM endothelium expresses PDGFRβ.** A) Representative sections of VMs and control neonatal dermis co-stained for CD31 and PDGFRβ. Red open arrowheads mark CD31+/PDGFRβ- ECs, yellow arrowheads mark CD31-/PDGFR β+ mural cells. White arrowheads mark CD31+/PDGFR β+ cells. A, B) Boxed areas are enlarged to the right. V-vein, VC-VM channel. Scale bars—50μm. V-vein, VC-VM channel.
(TIF)

**S3 Fig. Specificity of NOTCH3 antibody.** Neonatal dermis stained for NOTCH3 (left) and neonatal dermis stained for NOTCH3 which was blocked by pretreatment with a NOTCH3 blocking peptide (right). A-artery, V-vein. Scale bar—50μm.
(TIF)

**S4 Fig. The fetal endothelium expressed NOTCH3.** A fetal tissue array was co-stained for NOTCH3 and CD31. Representative images from adrenal gland, gallbladder, and umbilical cord shown. Boxed areas are enlarged to the right. Yellow arrowheads mark VECADHERIN-/NOTCH3+ mural cells, and white arrowheads mark VECADHERIN+/NOTCH3+ ECs. Scale bars—50μm.
(TIF)

**S5 Fig. VM endothelium expresses CD146.** A) Representative sections of VMs and control neonatal dermis co-stained for VECADHERIN and CD146. White arrowheads mark VECADHERIN+/CD146+ cells. A, B) Boxed areas are enlarged to the right. V-vein, VC-VM channel. Scale bars—50μm. V-vein, VC-VM channel.
(TIF)

**S6 Fig.**
(TIF)

**S1 Table. List of antibodies, sources, and dilutions.**
(DOCX)

**S2 Table. Summary of NOTCH3 expression in ECs of fetal organs.**
(DOCX)

## Acknowledgments

The authors would like to thank Ms. Noa Shapiro-Franklin and Ms. Emma Iaconetti for review of the manuscript.

## Author Contributions

**Conceptualization:** Carrie J. Shawber, June K. Wu.

**Data curation:** Michael J. Schonning, Seung Koh, Ravi W. Sun, Andrew K. Edwards.

**Formal analysis:** Andrew K. Edwards, Carrie J. Shawber, June K. Wu.

**Funding acquisition:** Carrie J. Shawber, June K. Wu.

**Investigation:** June K. Wu.

**Methodology:** Carrie J. Shawber, June K. Wu.

**Supervision:** Gresham T. Richter, Carrie J. Shawber.

**Writing – original draft:** Michael J. Schonning.

**Writing – review & editing:** Ravi W. Sun, Gresham T. Richter, Andrew K. Edwards, Carrie J. Shawber, June K. Wu.

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
