## [Decision Letter · Decision Letter 0]

22 Mar 2021

PONE-D-21-04851

Venous malformation vessels are improperly specified and hyperproliferative

PLOS ONE

Dear Dr. Wu,

Thank you for submitting your manuscript to PLOS ONE. After careful consideration, we feel that it has merit but does not fully meet PLOS ONE’s publication criteria as it currently stands. Therefore, we invite you to submit a revised version of the manuscript that addresses the points raised during the review process.

We look forward to receiving your revised manuscript.

Kind regards,

Wenbin Tan

Academic Editor

PLOS ONE

Journal Requirements:

Please provide additional details regarding participant consent. In the ethics statement in the Methods and online submission information, please ensure that you have specified whether consent was informed.

In your Methods section, please provide additional information about the participant recruitment method and the demographic details of your participants. Please ensure you have provided sufficient details to replicate the analyses such as: a) the recruitment date range (month and year), b) a description of any inclusion/exclusion criteria that were applied to participant recruitment, c) a description of how participants were recruited, and d) descriptions of where participants were recruited and where the research took place.

In your ethics statement in the manuscript, please ensure that you have discussed whether all control samples were fully anonymized before you accessed them and/or whether the IRB or ethics committee waived the requirement for informed consent. Please also provide the source of the foreskins.

Please note that PLOS does not permit references to “data not shown.” Authors should provide the relevant data within the manuscript, the Supporting Information files, or in a public repository. If the data are not a core part of the research study being presented, we ask that authors remove any references to these data.

Please ensure you have discussed any potential limitations of your study in the Discussion.

Thank you for including your ethics statement on the online submission form: "This study was approved by Columbia University’s IRB (AAAA9976). Each patient’s (orparents/legal guardian) whose tissues were used in this study signed a written consentform for the use of their tissues. Control neonatal foreskins were anonymouslycollected and no patients (or parents/legal guardian) signed a consent. The waiver forconsent was approved for controls tissues in the IRB.".  To help ensure that the wording of your manuscript is suitable for publication, would you please also add this statement at the beginning of the Methods section of your manuscript file.

8. We note that the grant information you provided in the ‘Funding Information’ and ‘Financial Disclosure’ sections do not match.

Review Comments to the Author

Reviewer #1: The authors provide a comprehensive characterization of gene expression and morphology of endothelial cells in venous malformations. Overall it is a well organized and well written manuscript that makes a strong argument for their stated goals. Most of my recommendations are minor grammatical corrections and organizational suggestions.

Abstract:

Ln 29 malformation WITH a prevalence...

Introduction:

Ln 53 -this sentence dangles and is probably not relevant

Lns 59-61 sentence probably irrelevant for an introduction

Ln 63 not new paragraph

Ln 73-5 probably move to discussion

Ln 77 n=16 is results, not introduction

Ln79-84 Again, these are results and this might better serve as the conclusion in the discussion

Materials:

Lns 88, 94, 95,117 n=x, again, results, not methods

Results:

Ln138 CD31 listed before VECADHERIN, subsequent results should follow same order. VEGFR2 should also be added to this list since it is covered later in the same paragraph

Ln 162 DLL4 discussed before EPHRINB2 so change order here.

Ln 168 Title say proteins decreased but EPHB4 actually increased. Please clarify

Discussion:

Lns 301-3 consider flipping sentence order

Like I said, trivial....

Reviewer #2: The manuscript by Schonning et al. aims at characterizing features of venous malformations endothelial and mural cell. The authors found that VMEC expressed arterial markers and was immature and hyperproliferative. The study is interesting to define the abnormal features of venous malformations vessels. However, there are still several concerns.

1. The immunofluorescent staining images have a poor quality and some are controversial with the quantitation. For example, in Figure 2B, the EPHB4 staining are relative lower in VM compared with control, which is not consist with the quantitation in figure 2D. Similar results were found in Figure 5A, Figure 7A et al.

2. No PDGFRβ staining were shown in Figure 5.

3. The study enrolled 16 VM specimens in all, but different numbers of specimens were used to observe different proteins expression. For different proteins, how did the authors select samples.

4. Author collected lesions from different locations, were there any differences from VMECs derived from different anatomical locations?

5. What are the differences of CD31 and VECADHERIN in labeling ECs in different samples? Were different EC markers used in different samples or all samples were stained with both markers? (CD31: Figure 1-3 and Figure 4A, VECADHERIN: Figure 4B and Figure 5-7).

---

## [Author Response · Author response to Decision Letter 0]

21 Apr 2021

We here submit our revised manuscript entitled, “Venous malformation vessels are improperly specified and hyperproliferative” for consideration as a Research Article in PLOS ONE. We thank the Reviewers for recognizing that it is “interesting” and “a comprehensive characterization of gene expression and morphology of endothelial cells in venous malformations, and that “it is a well-organized and well written manuscript that makes a strong argument for their stated goals”. We greatly appreciate the thoughtful Reviewers’ comments, which provided important guidance for the revised manuscript. We have included additional data and revised the manuscript to address reviewers’ concerns. Specific changes are described below. 

In response to Reviewer 1, we have addressed in blue the following minor grammatical errors and organizational suggestions:

1) In line 53 (now line 49), the sentence has been modified to reads: “Venous malformations (VMs) are slow-flow malformations of the venous vasculature.”

2) We have removed the sentence in lines 59-61.

3) At line 63 (now lines 55-56), we have merged the two paragraphs into a single paragraph.

4) Lines 73-75 probably move to discussion.

We have amended this sentence (now line 65) to read: “However, these studies are limited in scope and observational.”

6) Line 77 n=16 is results, not introduction.

n=16 was removed from the introduction and added that to the Results section (now line 123).

7) Lines 79-84 Again, these are results and this might better serve as the conclusion in the discussion

This sentence was moved to a new Conclusion section (now line 367).

8) Lines 88, 94, 95,117 n=x, again, results, not methods

Line 88: n=16 was removed. Line 94-95: The sentence was moved to the beginning of results section, under “Tissue Characteristics” (now lines 123-125). Line 117 was moved to the respective sections describing results of VECADHERIN and CD31 staining (now line 134-135), and NOTCH3 staining (now line 218-219) beginning of results section, under “Tissue Characteristics”.

9) Lines 138 CD31 listed before VECADHERIN, subsequent results should follow same order. VEGFR2 

 should also be added to this list since it is covered later in the same paragraph

The order of VECADHERIN and CD31 has been changed to be consistent throughout the paragraph. The sentence now reads: “To evaluate the tight and adherens junctions, VM specimens were stained for VECADHERIN, CD31 and VEGFR2” (lines 132-133).

10) Lines 162 DLL4 discussed before EPHRINB2 so change order here.

The order was corrected (now line 159).

11) Lines 168 Title say proteins decreased but EPHB4 actually increased. Please clarify

The title now reads: Venous endothelial proteins were altered in VM endothelium (now line 165).

12) Lines 301-3 consider flipping sentence order

The sentence order was switched (now lines 300-303).

We have made the following changes highlighted in blue in response to the Reviewer 2’s comments:

1) The immunofluorescent staining images have a poor quality and some are controversial with the quantitation. For example, in Figure 2B, the EPHB4 staining are relative lower in VM compared with control, which is not consist with the quantitation in figure 2D. Similar results were found in Figure 5A, Figure 7A et al.

We believe the poor quality was secondary to the pdf rendering when uploading to the journal. For the revision, we have used the PACE tool to generate the new figures according to journal guidelines.

As for the concern of figure images being consistent with the quantification, we have made the following changes:

The control in Figure 2B was replaced as the autofluorescence from the red blood cells in the original figure made the EPHB4 expression difficult to interpret. The VM image was also replaced to be more representative of the quantification. 

In Figure 5, we agree that there is no difference in the expression of NOTCH3 in the perivascular cells. However, there is difference in the expression of NOTCH3 in the endothelium in VM tissues which was not observed in the control tissues. We have clarified this in in the results section of the manuscript, lines 216-218.

In Figure 7, an alternative image that demonstrates an increase in the Ki67+ ECs lining the VM vessels is provided.

2) No PDGFRβ staining were shown in Figure 5.

A new Supplemental Figure 3 has been added that shows the PDGFRβ expression in control and VM tissues. 

3. The study enrolled 16 VM specimens in all, but different numbers of specimens were used to observe different proteins expression. For different proteins, how did the authors select samples.

This was a study conducted over multiple years during which VM specimen collection continued allowing for the increased number of samples as the study progressed. During the analysis, if we found that a smaller sample size was sufficient to achieve significance, we did not repeat the analysis to the complete set. 

4. Author collected lesions from different locations, were there any differences from VMECs derived from different anatomical locations?

Our VM samples came from a combination of head and neck, trunk and extremities. However, those numbers were too small for us to perform sub-analysis on similarities and differences of VMs from different anatomic locations.

5. What are the differences of CD31 and VECADHERIN in labeling ECs in different samples? 

We did an association analysis looking at VECAD and CD31 expression in VM samples (new Supplemental Figure 2). This analysis with Pearson correlation test showed weak association (R2 of 0.22). 

Were different EC markers used in different samples or all samples were stained with both markers? (CD31: Figure 1-3 and Figure 4A, VECADHERIN: Figure 4B and Figure 5-7).

All samples were stained for both CD31 and VECADHERIN. However, double staining was limited as several of the primary antibodies were generated in the same host as either the CD31 or VECADHERIN antibodies. For example, VEGFR2 and VECADHERIN were both made in goat, and thus, VEGFR2 was co-stained with CD31 (Figure 1B). Similarly, COUP-TFII was made in mouse (same as CD31), so COUPTFII was co-stained with VECADHERIN (Figure 2A).

We have also addressed the Journal Requirements. We have confirmed that our manuscript meets PLOS ONE’s requirements and figures were formatted using PACE. We have provided the required information on participant recruitment and consent, and well as all other additional information as requested in the Methods section. We have revised our ethics statement in the online submission form and added this statement to our Methods section. This required information includes the following: 

A new section titled “Patient recruitment and enrollment” in the Methods. 

Replacement of “data not shown” with a new Supplemental Figure 2. 

Addition of a discussion of potential limitations in a new last paragraph Discussion section.

The grant information in both “‘Funding Information’ and ‘Financial Disclosure’ sections have been reconciled to match, and we confirm that grant numbers are correct. The new Financial Statement should now read: “Funding: Funding was provided by DOD W81XWH1910266 (CJS) and DOD W81XWH1910267 (JKW). The funders had no role in the study design, data collection and analysis, decision to publish, or preparation of the manuscript. These studies used the resources of the Herbert Irving Comprehensive Cancer Center Pathology Shared Resources funded in part through Center Grant P30CA013696.

We appreciate your consideration of our revised manuscript.

---

## [Decision Letter · Decision Letter 1]

14 May 2021

Venous malformation vessels are improperly specified and hyperproliferative

PONE-D-21-04851R1

Dear Dr. Wu,

We’re pleased to inform you that your manuscript has been judged scientifically suitable for publication and will be formally accepted for publication once it meets all outstanding technical requirements.

Kind regards,

Wenbin Tan

Academic Editor

PLOS ONE

Reviewers' comments:

Reviewer #1: (No Response)

Reviewer #2: The authors have addressed all the comments and current version meets the requirements of publication.

---

## [Editor Report · Acceptance letter]

19 May 2021

PONE-D-21-04851R1 

Venous malformation vessels are improperly specified and hyperproliferative 

Dear Dr. Wu:

I'm pleased to inform you that your manuscript has been deemed suitable for publication in PLOS ONE. Congratulations! Your manuscript is now with our production department. 

Kind regards, 

on behalf of

Dr. Wenbin Tan 

Academic Editor

PLOS ONE